# Impact of a Dexmedetomidine Intravenous Infusion in Septic Dogs: Preliminary Study

**DOI:** 10.3390/ani14060892

**Published:** 2024-03-14

**Authors:** Chiara Di Franco, Søren Boysen, Iacopo Vannozzi, Angela Briganti

**Affiliations:** 1Department of Veterinary Sciences, University of Pisa, 56124 Pisa, Italy; iacopo.vannozzi@unipi.it (I.V.); angela.briganti@unipi.it (A.B.); 2Department of Veterinary Clinical and Diagnostic Sciences, Faculty of Veterinary Medicine, University of Calgary, Calgary, AB T2N 4Z6, Canada; srboysen@ucalgary.ca

**Keywords:** dexmedetomidine, dog, sepsis, CRI, SOFA, qSOFA

## Abstract

**Simple Summary:**

Sepsis is a widespread concern in human and veterinary medicine. Over the past several years, numerous strategies have been implemented to try to combat it; however, mortality remains high in all species. The use of dexmedetomidine, an alpha-two agonist drug with sedative and analgesic properties, is gaining popularity as a synergistic treatment strategy in the management of sepsis, thanks to its anti-apoptotic and neuroprotective properties and its ability to preserve hemodynamic function. We therefore hypothesized that a continuous rate infusion of dexmedetomidine in septic patients, undergoing emergency surgery, could decrease the requirement for vasopressors. The results of this study show that an infusion of 1 mcg/kg/h of dexmedetomidine decreases intraoperative vasopressor use and improves 28-day mortality. However, given the small sample size, a larger prospective study should be undertaken to confirm these findings.

**Abstract:**

The purpose of this study was to determine if a continuous rate infusion (CRI) of dexmedetomidine decreases vasopressor requirements in septic dogs undergoing surgery. Vital parameters, sequential organ failure assessment (SOFA) score, vasopressor requirement, and 28-day mortality were recorded. Dogs were randomly divided into two groups: a dexmedetomidine (DEX) (1 mcg/kg/h) group and a control group (NaCl), which received an equivalent CRI of NaCl. Dogs were premedicated with fentanyl 5 mcg/kg IV, induced with propofol, and maintained with sevoflurane and a variable rate fentanyl infusion. DEX or NaCl infusions were started 10 min prior to induction. Fluid-responsive hypotensive patients received repeated Ringer’s lactate boluses (2 mL/kg) until stable or they were no longer fluid-responsive. Patients that remained hypotensive following fluid boluses received norepinephrine at a starting dose of 0.05 mcg/kg/min, with increases of 0.05 mcg/kg/min. Rescue adrenaline boluses were administered (0.001 mg/kg) if normotension was not achieved within 30 min of starting norepinephrine. The NaCl group received a significantly higher dose of norepinephrine (0.8, 0.4–2 mcg/kg/min) than the DEX group (0.12, 0–0.86 mcg/kg/min). Mortality was statistically lower in the DEX group (1/10) vs. the NaCl group (5/6). Results of this study suggest that a 1 mcg/kg/h CRI of dexmedetomidine decreases the demand for intraoperative vasopressors and may improve survival in septic dogs.

## 1. Introduction

Sepsis, defined as life-threatening organ dysfunction caused by a dysregulated host response to infection [1], is a common emergency in both human and veterinary medicine. Unfortunately, the mortality rate remains high [2,3,4,5]. Recent experimental animal models and human clinical research suggests a possible protective effect of dexmedetomidine administration to septic subjects. Studies in rats demonstrate that dexmedetomidine has an inhibitory effect on the inflammatory response, lung protective effects, and reduces mortality rates in experimental sepsis models [6,7]. Furthermore, it has been demonstrated in human medicine that dexmedetomidine may reduce catecholamine requirements in patients with septic shock in the ICU setting [8,9]. Furthermore, it has also been shown that dexmedetomidine administration is associated with improved in-hospital survival in critically ill human patients with sepsis-associated acute kidney injury [10] and reduces the length of ICU stay in patients with sepsis on mechanical ventilation [11]. Despite the potential benefits of dexmedetomidine administration in the human ICU setting, in veterinary medicine, there is a lack of clinical research regarding its use in patients with sepsis. The aim of this study was to evaluate vasopressor requirements in septic dogs receiving a continuous infusion (CRI) of dexmedetomidine. We hypothesize that the administration of a dexmedetomidine CRI to septic dogs undergoing emergent surgery will decrease vasopressor requirements compared to a control group receiving normal saline solution.

## 2. Materials and Methods

Animal ethics approval (auth.n.24/2020) and client consent was obtained for all enrolled patients. Septic dogs admitted to the Intensive Care Unit (ICU) of the Veterinary Teaching Hospital University of Pisa that underwent emergency surgery for septic peritonitis were prospectively enrolled in a blind randomized fashion. Sepsis was defined as positive culture and/or presence of intracellular bacteria, and at least 2 out of 4 SIRS criteria [12]. Blood work (complete blood count, biochemistry panel, and blood gas analysis) was performed at arrival. Shock index (SI), sequential organ failure assessment (SOFA) and abbreviated SOFA (qSOFA) scores were calculated upon ICU admission. Vital parameters including heart rate (HR), temperature, respiratory rate, oscillometric arterial blood pressure (SAP, MAP), (PetTrust, BioCare, Aster Electrical Co., Ltd., Taoyuan, Taiwan), intra-operative vasopressor requirements, hospital discharge and the 28-day mortality rate were also recorded. Dogs were randomly divided into two perioperative groups using an online software program (https://www.random.org, accessed on 4 May 2021): a dexmedetomidine (DEX) group that received a constant rate infusion (CRI) of dexmedetomidine at 1 mcg/kg/h (Dextroquillan 0.5 mg/mL, Fatro, Ozzano dell’Emilia, Italy) and a control group (NaCl) that received an equivalent volume CRI of NaCl. All patients were premedicated with fentanyl 5 mcg/kg IV (Fentadon 50 mcg/mL, Eurovet Animal Health B.V, Dechra Veterinary Products, Turin, Italy) induced with propofol (Proposure, 10 mg/mL, Boehringer Ingelheim, Animal Health, Noventana, Italy) to effect and maintained with sevoflurane (SevoFlo, Zoetis, Zaventem, Belgium) at an end-tidal concentration (FE’Sevo) of 2.0–2.3% delivered in oxygen–air at an inspired oxygen fraction (FiO_2_) of 70%. Following anaesthetic induction, an IV catheter (18 or 20 G—Delta Med, Viadana, Italy) was placed in one of the two dorsal pedal arteries for continuous arterial blood pressure monitoring. The infusion (DEX or NaCl) was started 10 min prior to anaesthetic induction. A variable rate analgesic fentanyl infusion (3–20 mcg/kg/h) and a CRI of fluids (Ringer lactate) at 2 mL/kg/h was provided to all dogs while under general anaesthesia. Following anaesthetic induction, any episodes of arterial hypotension (MAP < 60 mmHg) were initially treated with a bolus of Ringer’s lactate (2 mL/kg IV). Patients that remained hypotensive and failed to respond to fluid bolus therapy, defined as an increase in arterial blood pressure or decrease in heart rate by at least 10% of pre-bolus values, were started on a variable rate IV infusion of norepinephrine 0.05 mcg/kg/min with incremental doses of 0.05 mcg/kg/min until normotension was restored. Patients that showed a response to fluid bolus therapy but remained hypotensive following two boluses (4 mL/kg total), were also started on a variable rate norepinephrine CRI with continued fluid boluses, until normotension was achieved (MAP > 60 mmHg). Fluid boluses were continued if the patient showed evidence of fluid responsiveness and discontinued if the patient failed to show fluid responsiveness and/or normotension was achieved. Patients that achieved normotension (MAP > 60 mmHg) within two IV boluses were not started on norepinephrine. If normotension was not achieved within 30 min of starting norepinephrine, rescue epinephrine boluses were administered (0.001 mg/kg). The attending anaesthetist was blinded to the study CRI protocol administered. Anaesthesia monitoring, as per the standard of care at the Veterinary Teaching Hospital, included recording of HR, invasive blood pressure, capillary refill time (CRT), ECG, EtCO_2_, and end FE’Sevo (Carescape B40, GE Healthcare, Bensalem, PA, USA) every 5 min. Intra-operative anaesthetic requirements were adjusted at the discretion of the attending anaesthetist. Dexmedetomidine and NaCl CRI’s were continued 24 h post-anaesthesia.

### Statistical Analysis

Considering a mean dosage of norepinephrine of 0.44 ± 0.19 mcg/kg/h [13] with an α error of 0.05 and β error of 0.2, in order to assess a decrease of 40% in the mean dosage of norepinephrine for the DEX group, a minimum requirement of 8 dogs in each group was calculated.

A D’Agostino and Pearson normality test was used to check data for normalcy. The two study groups were compared using a Student T or Mann–Witney test for parametric or non-parametric variables, respectively. Chi square analysis was performed to evaluate the number of dogs that received norepinephrine and/or epinephrine, and to compare mortality between the 2 groups. A one-way ANOVA was used to evaluate intraoperative MAP values at specific time points within each group, while a two-way ANOVA was used to compare all the MAP values of the 2 groups. *p* values < 0.05 were considered significant. GraphPad Prism 9 (GraphPad Software (Prism 10.2), LLC, Boston, MA, USA) was used to perform statistical analysis.

## 3. Results

Eighteen dogs were enrolled, ten in the DEX group, eight in the NaCl group. Surgical procedures in the DEX group included: two ovariohysterectomies for pyometra, one splenectomy, one laparotomy for periprostatic abscess, one gastric foreign body with perforation, and five intestinal foreign bodies with perforation; for the NaCl group, there were: two ovariohysterectomies for pyometra, one gastric dilatation volvulus (GDV), three intestinal foreign bodies with perforation, one intussusception and one volvulus. The mean age was 7.2 ± 4 years in the DEX group, and 6.2 ± 4 years in the NaCl group. The mean weight was 18 ± 11 kg in the DEX group, and 25 ± 10 kg in the NaCl group. There was no statistical difference between the groups for age (*p* = 0.59) or weight (*p* = 0.16). Pre-operatively, there was no statistical difference between the groups for vital signs, SI, HR, MAP, respiratory rate (RR), temperature, qSOFA, SOFA, lactate, albumin, creatinine, WBC and C Reactive protein (Table 1).

The mean propofol dosage was not statistically different (*p* = 0.67) between the two groups: 2.5 ± 1 mg/kg in the NaCl group and 2.7 ± 0.8 mg/kg in the DEX group.

For the intraoperative period, there were no statistical differences between the two groups for Ringer lactate administration (*p* = 0.82, 14 ± 10 mL/kg in the NaCl and 13 ± 6.5 mL/kg in the DEX group). In the DEX group 70% of dogs (7/10) received norepinephrine, while 100% of dogs (8/8) in the NaCl group received norepinephrine, which was not statistically different (*p* = 0.08). The median dosage of norepinephrine was significantly lower (*p* = 0.002) in the DEX group [0.12 (0–0.86) mcg/kg/min] compared to the NaCl group [0.8 (0.4–2) mcg/kg/min] (Figure 1). Two dogs in the DEX group, and four dogs in the NaCl group required an epinephrine bolus, which was not statistically different (*p* = 0.18). The median dose of fentanyl was not significantly different between the two groups (*p* = 0.15): 4.1 (3–9.9) mcg/kg/h in the DEX group and 5.5 (3–18.9) mcg/kg/h in the NaCl group. 

The mean arterial pressure during anaesthesia was not statistically different between the two groups (*p* = 0.17) (Figure 2). The 28-day mortality rate was statistically lower (*p* = 0.0034) in the DEX group (1/10) vs. the NaCl group (5/8). One dog in the DEX group and three dogs in the NaCl group were euthanized (all due to poor prognosis), while two dogs in the NaCl group died spontaneously.

The anaesthesia duration was not statistically different between the two groups: 124 ± 45 in the DEX group and 108 ± 49 min in the NaCl group; no correlation was found between the duration of anaesthesia and mortality.

The length of hospitalization between the two groups was not statistically different: 7 (4–10) days in the DEX group vs. 7 (2–14) days in the NaCl group (*p* = 0.14).

## 4. Discussion

The results of this study suggest a CRI administration of dexmedetomidine at a dosage of 1 mcg/kg/h in septic dogs may decrease intraoperative vasopressor use. 

The data analyzed also highlight a significant reduction in 28-day mortality between patients who received dexmedetomidine infusion in comparison to those who did not receive it. To the authors knowledge, this is the first clinical study in veterinary medicine to demonstrate a benefit of dexmedetomidine in septic dogs. 

These findings are in contrast with a recent canine clinical study that failed to show a mortality benefit or improved microcirculation when using dexmedetomidine compared with fentanyl in septic dogs with pyometra [14]. However, the prior study did find MAP was higher in dogs receiving a dexmedetomidine CRI during isoflurane anaesthesia compared to a fentanyl CRI. In the current study, we did not observe higher MAP values with dexmedetomidine infusion, but dexmedetomidine’s impact on blood pressure is likely a contributing factor to the lower norepinephrine dosages needed in the DEX group. However, it is difficult to directly compare the prior and current study given the differences in study populations (pyometra vs. any cause of septic peritonitis), the different perioperative anaesthetic regimes (isoflurane vs. sevoflurane and fentanyl compared to dexmedetomidine vs. saline compared to dexmedetomidine), differences in the study objectives (comparison of hemodynamic and microcirculatory variables vs. vasopressor requirements), and different dosages of dexmedetomidine (3 mcg/kg/h vs. 1 mcg/kg/h). It is unknown if a higher dose of dexmedetomidine in dogs of the current study would have influenced MAP and vasopressor requirements. The dosage used (1 mcg/kg/h) was based on prior reports of dogs undergoing surgery, which used dosages of 1 to 3 mcg/kg/h [14,15,16]. The 1 mcg/kg/h dose was chosen based on a study that suggests dexmedetomidine can contribute to a balanced and stable plane of postoperative analgesia for up to 24 h [17]. Further studies are required to determine if titrated dexmedetomidine dosages have an impact on mortality, MAP, and vasopressor requirements in septic dogs. 

The reduction in vasopressor requirements in septic dogs undergoing surgery is similar to what has been reported in human medicine and experimental animal sepsis models. A human ICU study showed that replacing propofol with dexmedetomidine provides a reduction in noradrenaline requirement in human septic patients [8]. In experimental septic ovine models in which sepsis is induced by inoculation of *Escherichia coli*, dexmedetomidine was found to decrease norepinephrine dosages, alleviate renal medullary hypoperfusion and hypoxia, and improve creatinine clearance over a 6 h interventional period when compared to sheep that received only norepinephrine [18]. The mechanism by which dexmedetomidine decreases the demand for vasoactive drugs in septic patients is likely multifactorial and not yet fully understood. It has been shown that alpha 2-agonists prevent downregulation and increase the sensitivity of alpha-1 adrenergic receptors, which may contribute to a decreased release of endogenous catecholamines in septic patients [19]. It is hypothesized that, by decreasing the levels of endogenous norepinephrine, the activity of exogenous norepinephrine on alpha1 receptors is increased. It is also possible that the presynaptic action of α-2 agonists can lead to an up-regulation of postsynaptic α-1 receptors by reducing the release of endogenous noradrenaline [20].

Results are mixed regarding mortality in human ICU patients receiving dexmedetomidine. Aso and colleagues report that dexmedetomidine is associated with reduced mortality in mechanically ventilated septic patients [21], which is similar to the mortality benefits of dexmedetomidine reported in septic patients by other authors [22,23]. By contrast, a meta-analysis by Abdelazeem et al., 2022, suggests that dexmedetomidine is not associated with reduced 28-day mortality, 90-day mortality, delirium-free days, ventilator-free days, changes in heart rate, or changes in mean arterial blood pressure when compared to the standard of care sedation in critically ill patients with sepsis [24]. A systematic review reports that dexmedetomidine may improve outcomes in patients with septic shock [25]. 

An advantage of dexmedetomidine may be the reduction in noradrenaline consumption: both these molecules cause vasoconstriction and bradycardia; however, dexmedetomidine reduces oxygen demand for the heart, and decreases cardiac output, maintaining renal microvascular blood flow [26,27,28]. Furthermore, the analgesic effect of dexmedetomidine can help reduce intraoperative sevoflurane requirements [29]. Recent in vitro studies have shown that dexmedetomidine has antioxidant effects and reduces oxidative stress and apoptosis in LPS-treated tubular epithelial cells [30]. Finally, it is reported that administration of high dose vasopressors (epinephrine and norepinephrine) during septic shock is associated with increased mortality rates of over 60% [31,32].

The current study has several limitations. The severity scores used may not reflect differences between groups, specifically with regards to sepsis. The qSOFA score for example, although associated with length of recovery and mortality in patients with sepsis undergoing pyometra [33], has been shown to be a poor predictor of mortality and has low sensitivity in identifying canine patients with severe sepsis and septic shock [34]. Although there did not appear to be a significant difference between groups regarding the severity of injury based on the SOFA, qSOFA, and SI, it is possible that a difference in management between the two groups could have contributed to a difference in the intra-operative requirements of vasopressors and overall mortality. The small sample size is also a limitation, which may have led to a false negative (type II error) in some of the statistical comparisons where a difference was not detected between groups, for example the severity of injury between the two groups. However, given that there was a significant difference detected in the intraoperative vasopressor use in septic dogs undergoing emergency surgery, and overall 28-day mortality, these results should be investigated further, controlling for confounding factors, as false positives (type I error) are unlikely to occur due to small sample size, instead being influenced by the selected p value (which accepts a 5% risk of a false positive result when set at 0.05, as chosen in the current study). Confounding factors that may have contributed to a type I error (false positive) include the fact that pre-operative management of patients, including the timing of antibiotics, the type of antibiotic, fluid volume, fluid type and fluid rates were not standardized or recorded. It is possible that the DEX group may have received earlier antibiotic administration, or a different fluid therapy plan, which subsequently decreased the requirement for vasopressor use during surgery. Hemodynamic optimization prior to induction of anaesthesia is indicated in septic patients [35]. However, there are no guidelines in veterinary medicine regarding the preoperative fluid approach. By contrast, in human medicine, the administration of 30 mL/kg of isotonic crystalloids in the first 3 h of ICU admission is often recommended, even if a personalized approach to patient management is suggested by others [36]. Monitoring for signs of fluid overload, multiple organ disfunction syndrome (MODS), evidence of acute kidney injury (AKI), or other signs of organ failure in the pre-, intra- and post-operative periods of the current study may have identified confounding factors which could influence findings. In the end, heart rate and mean arterial blood pressure were evaluated as hemodynamic indices, which lack sensitivity and specificity in evaluating fluid volume and fluid responsiveness; recording dynamic parameters such as cardiac output, pulse pressure variation and stroke volume variation, may have provided more precise fluid management intra-operatively, potentially decreasing the overall need for vasopressors. The dosages of vasopressor administered following surgery, while patients were hospitalized in the ICU, were not recorded, and this may have affected the 28-day mortality. However, as this was post-operative, any differences between groups would not have influenced the vasopressor doses administered intra-operatively. 

## 5. Conclusions

In conclusion, this study suggests that an infusion of dexmedetomidine at a dosage of 1 mcg/kg/h may contribute to a decreased requirement for intraoperative vasopressors in septic dogs undergoing emergency surgery. Due to uncontrolled confounding factors which may have influenced the results, claiming that an intraoperative infusion of dexmedetomidine reduces mortality in septic dogs at 28-days may be misleading. Nevertheless, the preliminary results of this study are encouraging and further studies with a larger number of cases are needed to confirm these findings.

## Figures and Tables

**Figure 1 animals-14-00892-f001:**
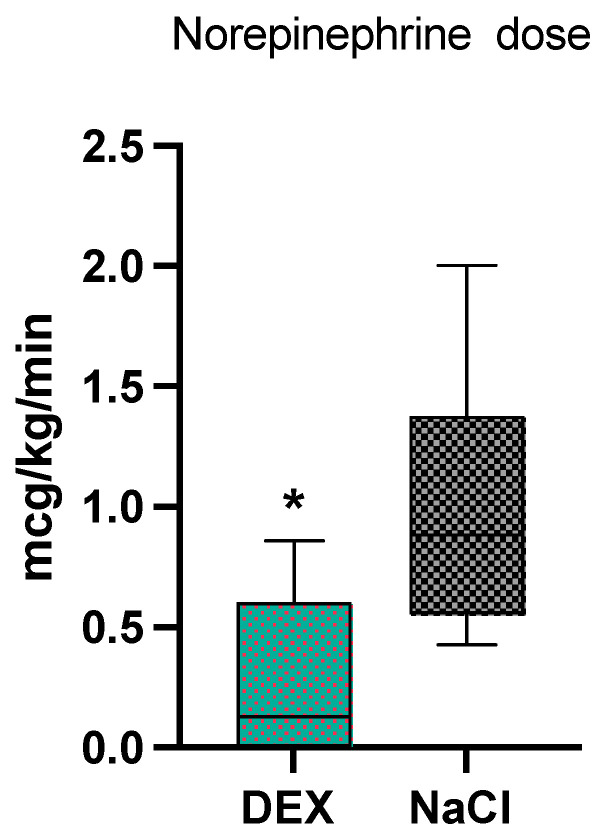
Norepinephrine median values and range in the 2 groups, dexmedetomidine (DEX) and normal saline (NaCl). * significantly different from NaCl group.

**Figure 2 animals-14-00892-f002:**
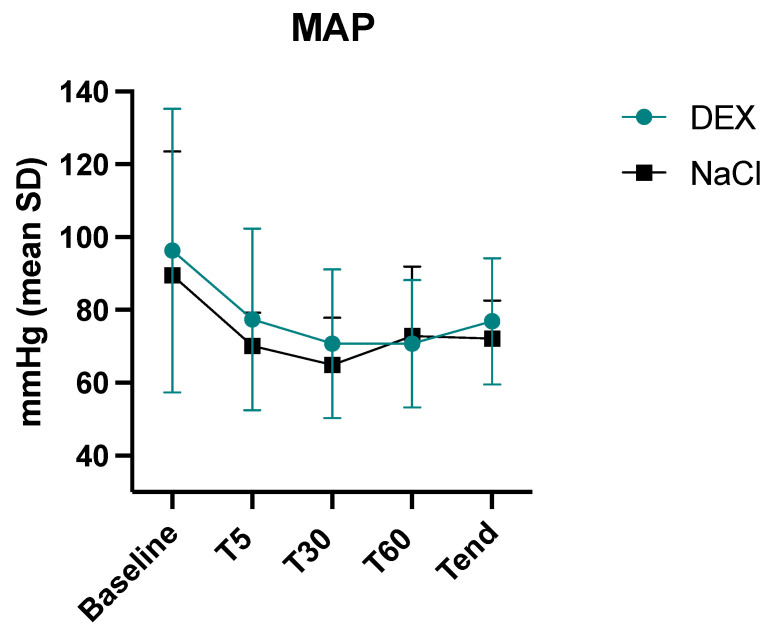
Intraoperative mean values and standard deviations of mean arterial pressure (MAP) in the 2 groups, dexmedetomidine (DEX) and normal saline (NaCl). Baseline, before anaesthesia; T5, five minutes after induction, T30, thirty minutes after induction, T60, sixty minutes after induction, Tend, last value before sevoflurane disconnection.

**Table 1 animals-14-00892-t001:** Comparison of preoperative clinical parameters between groups.

	DEX	NaCl	*p* Value
MAP (mmHg)	96 ± 39	90 ± 34	0.7
Heart rate (bpm)	153 ± 37	167 ± 34	0.47
Respiratory rate (apm)	34 ± 17	40 ± 13	0.42
Temperature (°C)	38 ± 0.9	37.7 ± 1.2	0.50
Shock index	1.36 ± 0.7	1.44 ± 0.6	0.82
SOFA	3.5 (1–13)	4 (0–6)	0.84
qSOFA	1 (0–2)	1 (0–3)	0.9
Lactate (mmol/L)	3 ± 1.8	2.9 ± 1.7	0.86
Albumin (g/dL)	2.7 ± 0.7	2.6 ± 1	0.7
Creatinine (mg/dL)	0.9 (0.5–2.6)	1.1 (0.5–9.8)	0.79
White blood cell count (K/µL)	12.6 (7–42.33)	9.6 (2.7–24.7)	0.15
RCP (mg/dL)	1.4 ± 1.1	1.6 ± 1.3	0.82

## Data Availability

Data supporting the results stated above can be sent to anyone requesting them from the authors.

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
