# Peer review of "Impact of a Dexmedetomidine Intravenous Infusion in Septic Dogs: Preliminary Study"

_animals, 2024, doi:10.3390/ani14060892_

Round 1

Reviewer 1 Report

Comments and Suggestions for Authors

Dear Authors,

Thank you for submitting this interesting study about the use of dexmedetomidine infusion in septic dogs. The manuscript is well written and easy to follow. I only have a couple of comments:

-line 29 and 95: typo for mcg/kg? should be mg/kg

-line 138: for the NaCl group should be 5/8 dogs and not 5/6 (unless 2 dogs were excluded from that group?) 

-did you have any lactate values post-op for those dogs? 

Author Response

Thank you for taking the time to review our study, and providing comments, which we have addressed below.

-line 29 and 95: typo for mcg/kg? should be mg/kg
The error in reporting the units has been corrected, thank you for point that out. 
-line 138: for the NaCl group should be 5/8 dogs and not 5/6 (unless 2 dogs were excluded from that group?) 
Thank you for the astute observation, the text has been corrected as correctly suggested. 
-did you have any lactate values post-op for those dogs? 
This is a great point, unfortunately we don’t have standardized lactate collection times for all patients at the completion of surgery, some were obtained immediately following surgery while others were not obtrained until several hours after returning to the ICU. For this reason we feel it is difficult to draw meaningful conclusions on the lactate values obtrain post-operativly. 

Reviewer 2 Report

Comments and Suggestions for Authors

The work is very interesting and could become significant, especially in cases of surgeries in septic patients. However, as highlighted by the authors, there are important limitations, such as the small group of treated patients and the lack of precise data on previous antibiotic treatments, as well as the timing in relation to surgery and the various causes of sepsis. Despite these limitations, the significant difference in 28-day mortality between patients administered dexmedetomidine and those receiving only lactated Ringer's infusion suggests a need for further study. 

Line 74: Was propofol administered “to achieve effect”? In this case, the effect of dexmedetomidine in reducing required dosage of propofol for induction should also be evaluated: the propofol has an important hypotensive action

Line 112-115: Were the various surgeries evenly distributed between the two groups? The difference in mortality, for example, between a pyometra intervention and one for enterectomy with septicemia, is already significant.

Line 235: replace “Finally” with “In the end”

Author Response

Thank you for reviewing the article and for your comments. We have tried to further emphasize the need for further prospective blinded studies on this topic. 

Line 74: Was propofol administered “to achieve effect”? In this case, the effect of dexmedetomidine in reducing required dosage of propofol for induction should also be evaluated: the propofol has an important hypotensive action
Thank you for this very intuitive comment, which we have addressed in the article. 

Line 112-115: Were the various surgeries evenly distributed between the two groups? The difference in mortality, for example, between a pyometra intervention and one for enterectomy with septicemia, is already significant. 
A very good point. We have added further detail regarding the diagnosis and reason for surgery for both groups to allow readers to draw conclusions and comparisons beyond just the injury severity scores.  

Line 235: replace “Finally” with “In the end” 
Corrected. 

Reviewer 3 Report

Comments and Suggestions for Authors

Review: Manuscript ID: animals-2858656Title: Impact of a dexmedetomidine intravenous infusion in septic dogs:preliminary studyAuthors: Chiara Di Franco *, Søren Boysen, Iacopo Vannozzi, Angela BrigantiVeterinary Clinical Studies

Many thanks for the opportunity to review this manuscript. The authors have evaluated the effect of a dexmedetomidine intravenous infusion in septic dogs and suggest that the effects on mortality are positive compared to standard norepinephrine.

In my opinion, the work is interesting and the question is worth investigating as the clinical problem is a very real and a daily challenge to clinicians dealing with septic patients requiring surgery.

My single main concern (and I feel that this may be a flaw that could result in refusal to publish this manuscript), is the small size of the groups studied. There is sufficient published data (cited in the manuscript) that the investigators could have used to estimate the number of animals required to reach an acceptably powered investigation. My concern is that the study is so underpowered that the chance of a type II error is so high that the data could in fact be misleading. I would suggest that what the authors may have done is a pilot study to justify a funding application to perform a properly powered study. If the investigators chose to resubmit this manuscript I would like to see a power calculation so that readers are afforded the opportunity to see this statistic and make a more informed assessment of the data presented.

Of less concern are the following:

1.     I think it important for the readers to be able to see in tabular form the causes of sepsis in the two groups evaluated. A clinical score (such as SOFA and qSOFA) do not tell the whole story of how ill the included dogs are and more than the SIRS criteria do. The very small groups and very diverse causes of sepsis included may make the groups incomparable.

2.     Line 182: e. coli should read Escherichia coli.

3.     Line 234: cofounding should read confounding

4.     Line 238: use of the ect is unhelpful.

5.     Line 240: ‘… is the fact the vasopressor …’ the English in this sentence requires correction.

All the figures and tables are useful and should be included.

The written English is of a good and publishable standard for an English language journal.

Comments on the Quality of English Language

The English language is good with only a few minor errors detected. 

Author Response

We thank the reviewer for taking the time to read and comment on our study.

My single main concern (and I feel that this may be a flaw that could result in refusal to publish this manuscript), is the small size of the groups studied. There is sufficient published data (cited in the manuscript) that the investigators could have used to estimate the number of animals required to reach an acceptably powered investigation. My concern is that the study is so underpowered that the chance of a type II error is so high that the data could in fact be misleading. I would suggest that what the authors may have done is a pilot study to justify a funding application to perform a properly powered study. If the investigators chose to resubmit this manuscript I would like to see a power calculation so that readers are afforded the opportunity to see this statistic and make a more informed assessment of the data presented.

We thank the reviewer for their comments and agree the sample size is a limitation of our study when we failed to detect statistical differences between groups due to a type II error – a false negative that failed to detect real differences due to small sample size. This is particularly pertinent in failing to detect differences between groups in the case of injury severity scores for example, which was highlighted as a limitation in the discussion.

However small sample size is less likely to lead to false positive findings (type I error), instead, being impacted by P values, which we set at 0.05 prior to conducting the research (as this is a commonly selected value and accepts a 5% risk of a type I error). We could have chosen a smaller P value to decrease the likelihood of a false positive type I error, however, we chose the commonly accepted p value prior to conducting the study, which cannot be changed retrospectively. Like the reviewer, we did not expect to see a statistical difference in mortality between groups with the p value chosen

Therefore, although we agree the study could serve as a pilot to calculate power for results that failed to reach significance between the groups (and determine the number of patients to detect a true effect when one is present, thereby decreasing the chance of a type II error), because we detected a difference in vasopressor use and mortality with the sample size used, power analysis cannot be used to detect a difference for these findings because a difference already exists.

We have added the power analysis, as the reviewer advised, to the statistics section to demonstrate the sample size we used to identify true differences between groups with regards to vasopressor use. As this was the main objective for the study, it was selected for power analysis. Other groups can use the results to determine the power and sample size needed to detect differences in other findings (such as injury severity score, propofol use etc), where we may have failed to detect statistical differences between groups that were truly present but failed to reach significance due to type II errors.

We have also expanded the discussion section on type I error and the p value selected. Finally, we have reworded the conclusion to emphasize the confounding factors between the groups and not the sample size that resulted in the statistical significance difference between the groups.

In conclusion, we feel the results of the current study are valid based on the p value chosen, power anlysis performed, methodology used, and for the reasons pointed out above, drawing from following article (Serdar CC, Cihan M, Yücel D, Serdar MA. Sample size, power and effect size revisited: simplified and practical approaches in pre-clinical, clinical and laboratory studies. Biochem Med (Zagreb). 2021;31(1):010502. doi:10.11613/BM.2021.010502).

Of less concern are the following:

  1. I think it important for the readers to be able to see in tabular form the causes of sepsis in the two groups evaluated. A clinical score (such as SOFA and qSOFA) do not tell the whole story of how ill the included dogs are and more than the SIRS criteria do. The very small groups and very diverse causes of sepsis included may make the groups incomparable.

Thank you, we added in the results section the distribution of the surgeries in the two groups.

  1. Line 182: e. coli should read Escherichia coli.

We corrected in the texts, as suggests.

  1. Line 234: cofounding should read confounding

Done

  1. Line 238: use of the ect is unhelpful.

we removed it

  1. Line 240: ‘… is the fact the vasopressor …’ the English in this sentence requires correction.

We corrected in the texts, thank you.

Round 2

Reviewer 2 Report

Comments and Suggestions for Authors

The introduction of statistical analysis (without significant difference) on the propofol dosage, and the specification of the interventions performed in the two groups, makes the work of greater scientific and clinical value. .

Although further studies are needed to confirm the usefulness of dexmedetomidine during sepsis, the work is of interest in being published.

Reviewer 3 Report

Comments and Suggestions for Authors

I am happy that the authors have adequately addressed my concerns. In my view, the manuscript may now be accepted for publication.